# Exploring the Associations Between School Climate and Mental Wellbeing: Insights from the MOVE12 Pilot Study in Norwegian Secondary Schools

**DOI:** 10.3390/ejihpe15040046

**Published:** 2025-03-26

**Authors:** Karoline Gulbrandsen Hansen, Svein Barene

**Affiliations:** 1Center for Studies of Educational Practice (SePU), University of Inland Norway, 2318 Hamar, Norway; karoline.g.hansen@inn.no; 2Department of Public Health and Sport Sciences, University of Inland Norway, 2406 Elverum, Norway

**Keywords:** school environment, mental health, physical education, health satisfaction

## Abstract

This study examined the association between school climate, defined by social and academic environments, and mental wellbeing among 446 first-year upper-secondary students in eastern Norway (ISRCTN10405415). As part of the MOVE12 pilot study conducted in February 2023, a cross-sectional online questionnaire targeted approximately 600 students from five schools offering diverse academic and vocational tracks. Mental wellbeing was assessed using the Warwick–Edinburgh Mental Wellbeing Scale (SWEMWBS, scale 7–35), and the data were analyzed with stepwise multiple linear regression. The mean mental wellbeing score was 24.5 ± 4.3, with significant gender differences (*p* < 0.05) but no variations between academic and vocational tracks. Self-efficacy was the strongest predictor of mental wellbeing (b = 0.236, *p* < 0.001), followed by health satisfaction (b = 0.179, *p* < 0.001), time spent with friends (b = 0.163, *p* < 0.001), social isolation (b = −0.162, *p* = 0.001), wellbeing in physical education (b = 0.129, *p* = 0.002), and classroom climate (b = 0.128, *p* = 0.007). These findings emphasize the critical role of self-efficacy, peer connections, and supportive classroom climates in promoting mental wellbeing. Addressing these elements of school climate can significantly enhance the mental health and overall outcomes of upper-secondary students.

## 1. Introduction

Mental wellbeing has been a central topic of research for decades, with numerous studies investigating the factors that contribute to this complex construct ([30]; [41]). Mental wellbeing concerns optimal experience and functioning, emphasizing the subjective experiences of happiness and life satisfaction ([17]; [68]). There are different theoretical approaches tied to wellbeing. Still, the term mental wellbeing can be referred to as a positive and sustainable mental state that allows individuals to thrive and flourish and focuses on both a feeling of happiness and the functionality of an individual ([28]; [43]; [54]). Developing positive mental wellbeing has long-term benefits, as adolescents with high wellbeing tend to perform better academically, have greater career success, and report higher life satisfaction in adulthood ([42]; [75]). Social relationships and physical health are key factors in fostering mental wellbeing ([48]; [62]; [68]), and school climate plays a particularly important role, as adolescents spend a significant portion of their daily lives in school ([5]; [34]).

School climate influences students’ mental wellbeing by shaping their social interactions, sense of belonging, and academic motivation. Self-Determination Theory (SDT) highlights the importance of autonomy, competence, and relatedness in fostering wellbeing ([67]). When students feel supported by teachers and peers, experience meaningful learning, and develop self-efficacy, they are more likely to engage in school positively. Conversely, a lack of support or inclusion can undermine motivation and wellbeing. The Socio-Ecological Model (SEM) further explains how individual wellbeing is influenced by interactions across multiple environmental levels ([56]). Supportive peer networks, inclusive extracurricular activities, and school-based mental health initiatives create an environment that fosters a sense of belonging and emotional security ([23]).

Research has shown that students’ perceptions of their school climate are closely linked to both health and psychosocial wellbeing ([14]; [27]; [76]; [79]), influencing self-esteem, depression, and anxiety ([2]; [40]; [61]). Students construct their identities through their experiences with their classroom climate and school climate by developing competencies and skills, understanding their social standing among peers, and enhancing their ability to self-regulate learning behaviors ([8]; [25]; [40]; [49]). The role of school climate is, therefore, pivotal in shaping students’ holistic development and mental wellbeing through extended socialization, fostering friendships, providing emotional support, and cultivating a sense of belonging ([27]; [36]; [74]; [76]).

Moreover, extensive scientific evidence highlights the positive effects of regular physical activity on mental wellbeing ([20]), primarily through the release of endorphins and serotonin, which play a critical role in mood regulation ([19]; [46]). Physical activity has also been shown to improve cognitive function, enhance academic performance, and support stress regulation ([81]). Given the increasing prevalence of sedentary behavior and declining physical activity levels among adolescents worldwide ([77]), integrating physical activity into daily routines could be a strategic approach to addressing the rising rates of mental health challenges in youth. Such an initiative could also positively impact school-related factors, including school climate, by fostering a sense of belonging, strengthening peer relationships, and promoting resilience ([4]; [13]).

This study aimed to explore the specific aspects of school climate that can contribute to students’ mental wellbeing. According to [78] ([78]), school climate can be described as a multidimensional construct that can be conceptualized into four areas: safety, community, academic environment, and institutional environment. Each of these areas plays a crucial role in supporting the functioning and flourishing of students ([78]). Within this framework, students’ mental wellbeing can be categorized into three main domains: (i) safety, (ii) community, and (iii) the academic environment. These areas are essential in helping students feel seen, heard, and valued as integral class members. The safety domain encompasses both physical safety and emotional safety, which are foundational for fostering a positive learning environment ([15]; [53]). The community domain involves the quality of relationships, sense of belonging, and school connectedness, which have been shown to significantly impact student engagement and mental wellbeing ([32]; [39]; [74]). Finally, the academic domain includes students’ self-efficacy and motivation, which are crucial for achieving academic success and personal growth. Self-efficacy, defined as the belief in one’s ability to influence events that affect one’s life, plays an important role in decision-making and perceptions of personal agency ([7]). Research suggests that young people’s perceived self-efficacy is shaped by contextual factors, including the school context ([18]; [39]).

Given that the different dimensions of school climate play a crucial role in meeting students’ psychosocial needs, it can be argued that when these needs are met, students are more likely to engage in learning and develop essential academic skills ([21]; [44]; [74]). Understanding the impact of mental wellbeing on both the present and the future lives of young people ([26]; [47]), this study aimed to investigate how school climate influences students’ mental wellbeing in upper-secondary schools.

Although some studies have identified a positive association between a supportive school climate and mental wellbeing in upper-secondary schools, most have primarily focused on younger students. Additionally, some of these studies have been conducted outside Western Europe ([60]), within sociocultural contexts that differ from the Norwegian setting. However, according to a recent Norwegian study, gender-dependent classroom climate influences emotional wellbeing ([65]). Given the unique developmental challenges faced by upper-secondary students and the need for context-specific insights, this study investigated the following research question:

‘How are dimensions of school climate, such as self-efficacy, classroom climate, social isolation, and wellbeing in physical education (PE), associated with the mental wellbeing of Norwegian upper secondary students, when accounting for variables such as gender, socioeconomic status (SES), study track, health satisfaction, and time spent with friends during leisure time?’

## 2. Materials and Methods

### 2.1. Study Design and Participants

This cross-sectional online questionnaire was conducted as a baseline assessment within a 12-week randomized controlled physical activity pilot study among Norwegian upper-secondary school students, where the study design has been previously presented in detail ([10]). In short, the intervention was conducted between January and May 2023. The target population included first-year students from three counties in eastern Norway. Efforts were made to ensure balanced representation from both academic and vocational study programs. Invitations were sent to 27 schools, and five were selected based on a stratified convenience sampling approach to achieve an approximately equal distribution of students from both categories. Among these, three were academic schools, and two were vocational. One vocational school, being twice the size of the others, contributed approximately 200 students, while the remaining four schools, three academic and one vocational, contributed about 100 students each. This selection ensured that the final sample included approximately 300 students from each study program, while also facilitating organization and implementation.

The inclusion criteria required students aged 16–17 years, while the exclusion criteria included disabilities or illnesses that posed participation or health risks. The study was approved by the Research Ethics Committee at Inland University Norway (21/01894) and registered in the International Standard Randomized Controlled Trial Number Register (ISRCTN10405415). Written informed consent was obtained from all participants.

The questionnaire incorporated validated scales to measure demographic background variables, mental wellbeing, and school-related factors, addressing the primary outcome variables of this study.

### 2.2. Demographic Background Variables

The demographic background variables included *gender*, *socioeconomic status (SES)*, *study track*, *time spent with friends*, and *health satisfaction*. *Gender* was recorded as ‘girl’, ‘boy’, or ‘other’, with ‘other’ being omitted from the statistical analyses due to the small sample size, ensuring confidentiality. *SES* was assessed using a question from The Norwegian Directorate of Health ([52]): ‘How easy or difficult is it for you to make ends meet on a daily basis with your household income?’. Responses were captured on a 7-point Likert scale ranging from ‘Very hard’ to ‘Very easy’, with an option for ‘I do not know’. *Study track* was categorized as either academic or vocational, with additional details provided for specific programs within those tracks. Social interaction was measured using a single item from the Oslo Social Support Scale ([52]), which assessed the *time spent with friends* on a 6-point Likert scale from ‘Almost daily’ to ‘Less often than every year’, including the option ‘I do not have any good friends’. Lastly, *health satisfaction* was measured using a single item from the Short Form Survey Instrument ([80]), asking, ‘How do you perceive your health status today?’. Responses were recorded on a 5-point Likert-scale ranging from 1 = ‘Bad’ to 5 = ‘Excellent’.

### 2.3. Mental Wellbeing

*Mental wellbeing* was assessed using the short version of the Warwick–Edinburgh Mental Wellbeing Scale (SWEMWBS), a validated tool particularly suited for Norwegian adolescents ([64]; [71]; [73]). This 7-item scale captures wellbeing on a 7-35 range and includes statements such as ‘I have been optimistic about the future’, ‘I have felt useful’, and ‘I have been thinking clearly’. Responses are rated on a 5-point Likert scale ranging from 1 = ‘Not at all’ to 5 = ‘All the time’. The scale demonstrated good internal consistency in this study, with a Cronbach’s α of 0.789.

### 2.4. Self-Efficacy

*Self-efficacy* was assessed using a scale specifically adapted for Norwegian students by [72] ([72]) based on [9] ([9]). The scale consists of four items that gauge students’ perceptions of their self-efficacy, with statements such as ‘I think I can handle the tasks I am given in class’ and ‘I give up if I find the task difficult’. Responses were recorded on a 5-point Likert scale ranging from 1 = ‘No, never’ to 5 = ‘Yes, always’. The scale demonstrated acceptable internal consistency in this study, with a Cronbach’s α of 0.722.

### 2.5. Classroom Climate and Social Isolation

*Classroom climate* was measured using a scale originally developed by [50] ([50]), which was later adapted and translated into Norwegian by [72] ([72]) and [55] ([55]). While the original scale comprised 13 items, this study focused on nine items specifically related to the classroom climate, yielding a Cronbach’s α of 0.847, indicating good internal consistency. The scale includes items such as ‘I can ask a classmate for help if there is something I do not understand’ and ‘The students in my class know each other well’, with responses rated on a 4-point Likert scale ranging from 1 = ‘Totally disagree’ to 4 = ‘Totally agree’.

To assess perceived *social isolation* at school, the study employed a modified version of the ‘Social skills rating system’ developed by [29] ([29]), also translated and adapted for Norwegian use by [72] ([72]). Originally containing 21 items across three factors, the scale was condensed to 5 items focusing on social isolation, including statements like ‘I feel sad at school’, ‘I am alone during recess’, and ‘I am excluded from the social relationships at school’. The five-item social isolation scale demonstrated acceptable reliability with a Cronbach’s α of 0.789.

### 2.6. Wellbeing in Physical Education

To assess *wellbeing in physical education (PE)*, we adapted the wellbeing in school scale from the Norwegian Ungdata survey ([6]) specifically for PE classes. The scale comprises five items, including statements such as ‘I enjoy physical education classes’ and ‘I feel that I fit in among the students in physical education’. Responses were measured on a 4-point Likert scale ranging from 1 = ‘Totally disagree’ to 4 = ‘Totally agree’. The scale demonstrated good internal consistency in this study, with a Cronbach’s α of 0.746.

### 2.7. Statistical Analysis

Descriptive item statistics were initially explored using STATA version 18 (StataCorp LP, College Station, TX, USA). Preliminary analysis included summary statistics of demographic data, followed by factor analyses to validate the constructs. Correlation analyses were then performed to examine the relationships between the scales and mental wellbeing.

All participants who provided informed consent and completed the questionnaire were included in the dataset. However, seven responses were excluded from the analyses because the participants identified as ‘Other’ in response to the gender identity question (Boy, Girl, Other). Due to the small sample size in this category, meaningful statistical comparisons were not feasible.

A stepwise multiple linear regression was conducted with mental wellbeing as the dependent variable. To ensure the validity of the results, we verified that the assumptions of normality, linearity, multicollinearity, and homoscedasticity were not violated. *Gender*, *study track*, and *SES* were entered at Step 1, explaining 6% of the variance in *mental wellbeing*. At Step 2, *time spent with friends* and *health satisfaction* were added, increasing the explained variance to 20%. Finally, at Step 3, the inclusion of the four school-related variables, *wellbeing in PE*, *self-efficacy*, *classroom climate*, and *social isolation*, enhanced the model, explaining 37% of the variance in *mental wellbeing*, F (9,436) = 29.5, *p* < 0.001. The beta coefficients were subsequently analyzed to assess the relative impact of each predictor, highlighting their significance to students’ *mental wellbeing*.

## 3. Results

### 3.1. Descriptive Statistics

Out of 739 eligible students across five schools, 446 (∼60%) participated in the online electronic questionnaire (see Table 1), comprising 247 girls and 199 boys. Of these, 253 students (∼57%, 140 girls, 113 boys) were enrolled in academic tracks, while 193 (∼43%, 107 girls, 86 boys) were in vocational tracks. No significant differences were observed between academic and vocational groups in terms of *SES*, *time spent with friends* outside school, or *mental wellbeing*. However, *health satisfaction* varied significantly, with students in the academic track and boys reporting higher satisfaction levels (Table 1).

### 3.2. Correlation and Multicollinearity

To explore associations between the dependent variable and the explanatory variables, a correlation analysis was conducted (Table 2). However, additional analyses using the variance inflation factor (VIF) indicated no multicollinearity among the independent variables, with all VIF values below 1.6.

### 3.3. School Climate and Mental Wellbeing

To systematically explore potential mediating relationships between the variables, a 3-step multiple regression analysis was used (Table 3). This approach is particularly useful for examining complex models, as it allows for stepwise identification of direct effects, mediators, and moderators ([12]; [35]). Our model explained 36.6% of the variance in mental wellbeing, which is considered acceptable.

As shown in Figure 1, the analysis revealed that all school climate variables significantly predicted mental wellbeing. *Self-efficacy* was the strongest predictor (b = 0.236, *p* < 0.001), followed by *social isolation* (b = −0.162, *p* = 0.001), *wellbeing in physical education (PE)* (b = 0.129, *p* = 0.002), and *classroom climate* (b = 0.128, *p* = 0.007). Additionally, Model 3 demonstrated that *health satisfaction* (b = 0.179, *p* < 0.001) and *time spent with friends* (b = 0.163, *p* < 0.001) remained significant predictors, highlighting the complex array of factors within the school environment that influence students’ mental wellbeing (Table 3).

Interestingly, *socioeconomic status (SES)* showed no significant association with mental wellbeing in our analysis (b = 0.026, *p* = 0.497). This suggests that SES has a limited influence on mental wellbeing in this context when other factors are considered.

## 4. Discussion

The aim of this study was to explore how students’ perceptions of their school climate could impact their self-perceived mental wellbeing. Our findings indicate that various dimensions of school climate, as well as both *health satisfaction* and *time spent with friends* in leisure time, are significant predictors of mental wellbeing, underscoring the importance of considering these factors in educational settings. The regression analysis revealed that *wellbeing in PE*, *self-efficacy*, *classroom climate*, and *social isolation* impact students’ mental wellbeing, even after controlling for *gender*, *study track*, *health satisfaction*, and *time spent with friends* during leisure time.

In our model, *self-efficacy* emerged as the most robust predictor of wellbeing (b = 0.236; *p* < 0.000). This suggests that students who perceive themselves as capable and competent are more likely to experience higher levels of mental wellbeing. This aligns with previous research that emphasizes the role of self-efficacy in fostering a positive self-image and enhancing mental health ([16]; [24]; [30]). Self-efficacy is a dynamic construct where the environment and personal behavior interact with thoughts, feelings, and beliefs ([7]). Therefore, the school environment is crucial in shaping students’ self-efficacy, influencing their motivation and learning ([33]). Research by [24] ([24]) underscores that students with higher scholastic self-efficacy tend to experience greater wellbeing ([24]), demonstrating the strong connection between perceived competence and psychological health. Nonetheless, [44] ([44]) highlights that the benefits of a positive school climate on student engagement are dependent on its effective implementation and its ability to enhance students’ wellbeing experiences ([44]). This suggests that while fostering a supportive school environment is essential, its actual impact on student wellbeing hinges on how well it translates into meaningful improvements in students’ daily experiences and perceptions.

Our findings suggest that social relationships, both within and outside school, significantly predict students’ mental wellbeing. *Classroom climate* (b = 0.128; *p* = 0.007) remained a significant predictor, even when controlling for *time spent with friends* outside school (b = 0.163; *p* < 0.001). Strong social relationships are crucial to mental wellbeing, as they provide emotional support, a sense of belonging, and opportunities for social interaction, all of which contribute to psychological adjustment ([78]). The classroom climate significantly impacts wellbeing, especially for young people who spend substantial time interacting with their peers and classmates during school days ([51]). A strong sense of belonging and social support within the classroom setting is closely associated with enhanced mental wellbeing, as students who feel seen, respected, and included tend to experience higher levels of wellbeing ([3]; [16]; [30]).

However, *classroom climate* alone did not explain the *gender* difference in mental wellbeing, as the effect remained significant (*p* < 0.001) after its inclusion. The gender disparity only disappeared after adding *self-efficacy*, *social isolation*, and *wellbeing in PE*, suggesting that these factors collectively account for the difference. Girls may experience lower self-efficacy, greater social isolation, or different PE experiences, which could contribute to their lower wellbeing. Unlike previous research indicating gender-specific effects of classroom climate ([65]), our study suggests that multiple school-related factors drive gender disparities in mental wellbeing. While our study did not directly examine body image satisfaction, social comparison, or academic stress, prior research has highlighted that adolescent girls often report lower body image satisfaction, greater vulnerability to social comparison, and higher academic stress related to societal expectations, factors that have been linked to self-esteem and overall wellbeing ([11]; [38]; [57]). These broader patterns may help contextualize the gender differences observed in our study, underscoring the need for further research into the complex interplay of psychological and social factors in adolescent mental health.

This aligns with attachment theory, which posits that consistent emotional support and a secure environment foster self-reliance and resilience ([1]; [78]), enabling young people to make autonomous decisions rather than being influenced by their surroundings. Schools characterized by high-quality relationships and a sense of connectedness not only boost academic achievements but also contribute to positive health outcomes ([74]; [78]), emphasizing the pivotal role of a supportive school environment in promoting holistic student development.

Our model indicates that *social isolation* may negatively impact students’ wellbeing (b = −0.162; *p* = 0.001). Social isolation is often linked to reduced wellbeing and can lead to feelings of loneliness, which may exacerbate existing mental health issues ([26]). Physical and emotional safety are integral components of a positive school climate, as they allow students to express themselves without fear of negative reactions. Students who feel unsafe, or experience social isolation or bullying, are likely at a higher risk of lower wellbeing ([26]).

Not surprisingly, the study demonstrated that perceived *wellbeing in physical education (PE)* classes significantly enhanced students’ overall mental wellbeing (b = 0.129; *p* = 0.002). This finding aligns with existing research indicating that physical education offers students crucial opportunities for physical activity, social interaction, and skill development, all of which are instrumental in enhancing self-esteem and mental health ([5]; [37]). The beneficial effects of physical activity on mental health are well-documented, with evidence indicating improvements in mood, reduction in anxiety levels, and enhanced cognitive function ([13]; [45]). It should be emphasized that physical activity represents a powerful tool for enhancing wellbeing and improving students’ quality of life ([19]). While this study does not assess the direct effects of physical activity, future research should further explore how structured school-based interventions can optimize its implementation as a mental health-promoting strategy in educational settings.

It is important to acknowledge that wellbeing in the school environment is shaped not only by individual experiences but also by broader institutional factors, including the school’s approach to and implementation of physical activity ([77]). While this study represents a baseline assessment conducted before the implementation of the physical activity intervention, prior research suggests that school policies, teacher attitudes, and the availability of structured physical activity opportunities significantly influence students’ engagement and wellbeing ([13]; [31]). Future research should examine how the implementation of structured physical activity programs impacts school climate and student wellbeing outcomes beyond individual perceptions of PE. These findings reinforce the critical role of well-structured PE programs in promoting mental wellbeing among students.

Unlike previous studies identifying SES as a key predictor of mental wellbeing ([59]; [63]), the significant association observed in Model 1 (*p* = 0.001) disappeared after adjusting for *spending time with friends* and *health satisfaction* in Model 2 (*p* = 0.068) and was further diminished in Model 3 (*p* = 0.497). This suggests that social relationships and perceived health mediate the SES-wellbeing link, consistent with research highlighting their protective role ([22]). Another possible explanation is the limited socioeconomic diversity in our sample, which may have reduced variability and obscured potential associations. Additionally, the findings align with the resource substitution hypothesis ([69]; [70]), which posits that strong social and environmental support can mitigate the negative effects of SES disparities. In this context, school-related factors, such as supportive peer and teacher relationships, likely played a compensatory role, buffering the impact of SES on wellbeing ([58]). These protective mechanisms may have weakened the direct association between SES and mental wellbeing, underscoring the importance of social resources in shaping adolescent mental health outcomes. Future research should examine these dynamics in more socioeconomically diverse samples and employ longitudinal designs to clarify how school environments and peer relationships mediate SES effects over time. This would help identify targeted interventions to reduce socioeconomic disparities in adolescent mental health.

The findings should be understood in the Norwegian context, where equal access to education, healthcare, and social support most likely helps reduce SES differences in mental wellbeing. Instead, *self-efficacy* emerged as the strongest positive predictor, followed by *time spent with friends* and *health satisfaction*, emphasizing the role of personal agency and social connectedness. Additionally, classroom climate ranked third, reinforcing the impact of inclusive, student-centered learning environments. In contrast, social isolation was a significant negative predictor, underscoring its harmful effects. These results suggest that in Norway, school climate and peer relationships are central to adolescent mental wellbeing, differing from societies where SES plays a stronger role due to greater economic inequality or competitive educational systems ([63]; [66]).

### 4.1. Strengths and Limitations

A strength of the study is the relatively robust data collection employing validated scales across key constructs such as *mental wellbeing*, *self-efficacy*, and *classroom climate*, which ensured accurate and dependable measurements. An additional strength is the 61% response rate from both academic and vocational programs enhancing the study’s generalizability. However, a potential weakness of the study is the use of self-reported data, which may lead to response bias due to socially desirable answering tendencies. Furthermore, while the five participating schools were geographically distributed across southeastern Norway, their location in mid-sized urban areas may limit the study’s representativeness. To improve the external validity, future research should strive to include a more diverse sample, incorporating schools from various regions, including rural areas.

### 4.2. Educational Strategies and Future Research Directions

Our findings underscore the importance of school climate in shaping students’ mental wellbeing. Self-Determination Theory (SDT) highlights that when students experience autonomy, competence, and relatedness, they are more likely to develop intrinsic motivation, emotional resilience, and overall wellbeing ([67]). The results indicate that positive teacher–student relationships, peer support, and engagement in school activities contribute to students’ sense of belonging and self-efficacy, supporting SDT’s emphasis on fulfilling these psychological needs. Similarly, the Socio-Ecological Model (SEM) illustrates how school climate interacts with individual and social factors to shape wellbeing ([56]). The classroom climate, peer relationships, and extracurricular participation play a crucial role in either mitigating or reinforcing SES-related disparities, social isolation, and differences in health satisfaction. Our findings suggest that a supportive school climate can buffer the negative effects of socioeconomic disadvantages by fostering positive interactions, meaningful engagement, and emotional security. These results highlight the need for school-based interventions that enhance peer integration, improve teacher–student relationships, and create inclusive learning environments.

Future research should further explore these mechanisms by examining how teacher–student dynamics contribute to self-efficacy, the impact of extracurricular activities on social connectedness, and the role of digital technology as both a risk factor and a support tool for mental wellbeing. Understanding these factors will provide valuable insights for designing evidence-based educational policies that promote students’ wellbeing in diverse learning contexts.

## 5. Conclusions

Our study reveals that various dimensions of school climate, including *self-efficacy*, *classroom climate*, and *wellbeing in physical education*, are significant predictors of students’ mental wellbeing. *Self-efficacy* emerged as the strongest predictor, indicating the critical role of perceived competence. Social relationships, both within and outside school, also play a vital role, while *social isolation* negatively impacts wellbeing. In contrast, the study suggests that neither *gender* nor *SES* has a substantial influence on shaping students’ mental health outcomes. These findings underscore the importance of a supportive school climate, particularly for upper-secondary students, in promoting mental health.

## Figures and Tables

**Figure 1 ejihpe-15-00046-f001:**
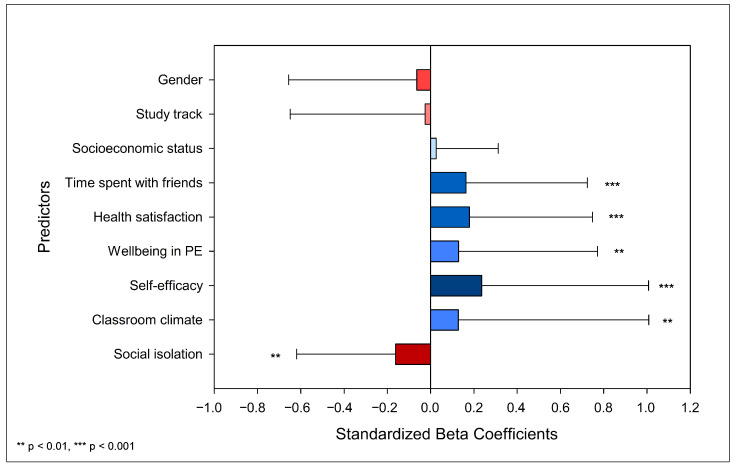
Predictors of mental wellbeing among Norwegian upper-secondary school students (error bars represent 95% CI).

**Table 1 ejihpe-15-00046-t001:** Descriptive characteristics of the study population (n = 446) at baseline, stratified by gender and study track. Data are reported as mean (SD).

Characteristics	Gender			Study Track		
	Boys	Girls		Academic	Vocational	
	(n = 199)	(n = 247)	*p*-Value	(n = 253)	(n = 193)	*p*-Value
Socioeconomic status (1–7)	5.1 ± 1.2	5.1 ± 1.3	0.785	5.2 ± 1.1	5.1 ± 1.4	0.410
Time spent with friends (1–6)	5.5 ± 0.9	5.3 ± 0.9	0.140	5.4 ± 0.9	5.4 ± 0.8	0.974
Health satisfaction (1–5)	3.6 ± 1.0	3.2 ± 0.8	**0.000**	3.5 ± 0.9	3.2 ± 0.9	**0.000**
Mental wellbeing (7–35)	25.4 ± 4.5	23.8 ± 4.0	**0.000**	24.8 ± 4.0	24.2 ± 4.6	0.144
Self-efficacy (1–5)	3.9 ± 0.6	3.7 ± 0.6	**0.000**	3.8 ± 0.6	3.7 ± 0.6	0.357
Classroom climate (1–4)	3.2 ± 0.5	3.2 ± 0.5	0.336	3.2 ± 0.5	3.2 ± 0.6	0.389
Wellbeing in PE (1–4)	3.1 ± 0.7	3.1 ± 0.6	0.414	3.1 ± 0.7	3.1 ± 0.7	0.420
Social isolation (1–5)	1.5 ± 0.6	1.7 ± 0.6	**0.008**	1.6 ± 0.6	1.6 ± 0.7	0.592

**Table 2 ejihpe-15-00046-t002:** Pearson’s correlation matrix for all included variables (n = 446).

	Characteristics	1	2	3	4	5	6	7	8
1	Wellbeing (7–35)	-							
2	Socioeconomic status (1–7)		0.16 **						
3	Time spent with friends (1–6)			0.32 ***					
4	Health satisfaction (1–5)				0.37 ***				
5	Self-efficacy (1–5)					0.36 ***			
6	Classroom climate (1–4)						0.37 ***		
7	Wellbeing in PE (1–4)							0.32 ***	
8	Social isolation (1–5)								0.42 ***

*** *p* < 0.001, ** *p* < 0.01.

**Table 3 ejihpe-15-00046-t003:** Multiple regression analysis (forward stepwise) of predictors for mental wellbeing.

Model	Unstandardized		Standardized	t	*p*-Value	Adjusted
	Coefficient		Coefficient			R2
	B	St. Error	Beta			
(Constant)	24.974	1.198	−	20.85	0.000	0.057
Gender	−1.566	0.395	−0.183	−3.97	**0.000**	
Study track	−0.541	0.396	−0.063	−1.37	0.173	
Socioeconomic status	0.540	0.158	0.158	3.43	**0.001**	
(Constant)	14.833	1.614	−	9.19	0.000	0.198
Gender	−0.972	0.372	−0.113	−2.61	**0.009**	
Study track	−0.195	0.371	−0.023	−0.53	0.599	
Socioeconomic status	0.272	0.149	0.080	1.83	0.068	
Time spent with friends	1.082	0.223	0.218	4.85	**0.000**	
Health satisfaction	1.270	0.216	0.271	5.88	**0.000**	
(Constant)	7.830	2.350	−	3.33	0.001	0.366
Gender	−0.553	0.335	−0.064	−1.65	0.100	
Study track	−0.218	0.331	−0.025	−0.66	0.509	
Socioeconomic status	0.091	0.133	0.026	0.68	0.497	
Time spent with friends	0.810	0.203	0.163	3.99	**0.000**	
Health satisfaction	0.837	0.199	0.179	4.22	**0.000**	
Wellbeing in PE	0.810	0.262	0.129	3.09	**0.002**	
Self-efficacy	1.619	0.273	0.236	5.94	**0.000**	
Classroom climate	1.043	0.384	0.128	2.71	**0.007**	
Social isolation	−1.076	0.316	−0.162	−3.40	**0.001**	

## Data Availability

The raw data supporting the conclusions of this article will be made available by the authors on request.

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
