# Peer review of "Exploring the Associations Between School Climate and Mental Wellbeing: Insights from the MOVE12 Pilot Study in Norwegian Secondary Schools"

_ejihpe, 2025, doi:10.3390/ejihpe15040046_

Round 1
Reviewer 1 Report
Comments and Suggestions for Authors
Dear authors, thank you for the opportunity to work. The manuscript used concerns an important social issue. The one sent to me for work requires several corrections and additions within the individual sections:
Introduction:
-Please add information on the role of physical activity and its beneficial effect on the mental sphere and well-being in this section. It is not only the release of endorphins as a result of physical activity but also a number of effects related to relieving stress, improving self-esteem, self-acceptance, a sense of satisfaction or building social relationships.
Material and methods:
In this section, please add a graph illustrating the course of the research experiment, taking into account the individual phases of the experiment, group size, inclusion and exclusion criteria, etc.
-Please also add information on the method used to select the research group
-Please also add information on whether any questionnaires were rejected from the analysis and for what reason?results: -Please highlight statistically significant results in tables -Please present key results for the study in graphical form, which will certainly be more attractive to potential recipients
Author Response
Response to Reviewer 1
Overall comment:
Dear authors, thank you for the opportunity to work. The manuscript used concerns an important social issue. The one sent to me for work requires several corrections and additions within the individual sections:
Response:
Thank you for taking the time to help strengthen our manuscript and for your generally positive feedback. We really appreciate it!
Comment 1:
Introduction:
-Please add information on the role of physical activity and its beneficial effect on the mental sphere and well-being in this section. It is not only the release of endorphins as a result of physical activity but also a number of effects related to relieving stress, improving self-esteem, self-acceptance, a sense of satisfaction or building social relationships.
Response 1:
Thank you for your valuable comment. In response, we have included information on the beneficial effects of physical activity on mental wellbeing through the following formulation:
L57-68: “Moreover, extensive scientific evidence highlights the positive effects of regular physical activity on mental wellbeing (Cheng et al., 2025), primarily through the release of endorphins and serotonin, which play a critical role in mood regulation (Mandolesi et al., 2018; Cadenas-Sanchez et al., 2021). Physical activity has also been shown to improve cognitive function, enhance academic performance, and support stress regulation (Zarazaga-Peláez et al., 2024). Given the increasing prevalence of sedentary behavior and declining physical activity levels among adolescents worldwide (van Sluijs et al., 2021), integrating physical activity into daily routines could be a strategic approach to addressing the rising rates of mental health challenges in youth. Such an initiative could also positively impact school-related factors, including school climate, by fostering a sense of belonging, strengthening peer relationships, and promoting resilience (Allen et al., 2022; Biddle and Asare, 2011).”
Comment 2:
Material and methods:
In this section, please add a graph illustrating the course of the research experiment, taking into account the individual phases of the experiment, group size, inclusion and exclusion criteria, etc.
Response 2:
Thank you for your request for a more detailed description of the physical activity intervention study design. However, we respectfully maintain that this information is not directly relevant to the present study, which focuses solely on a baseline assessment, a cross-sectional survey conducted at a single measurement point prior to the start of the 12-week intervention (January–May 2023). The potential effects of the intervention are being examined in a separate study, currently under peer review in another journal, where the study design and methodology are thoroughly documented.
That said, we recognize the relevance of your comment and have revised the manuscript accordingly. Specifically, we have included a reference to a previously published protocol article that provides a comprehensive overview of the intervention study design. To enhance clarity, we have added the following description at the beginning of subsection 2.1:
L109-122: “This cross-sectional online questionnaire was conducted as a baseline assessment within a 12-week randomized controlled physical activity pilot study among Norwegian upper secondary school students, where the study design has been previously presented in detail (Barene et al., 2025). In short, the intervention was conducted between January and May 2023. The target population included first-year students from three counties in Eastern Norway. Efforts were made to ensure balanced representation from both academic and vocational study programs. Invitations were sent to 27 schools, and five were selected based on a stratified convenience sampling approach to achieve an approximately equal distribution of students from both categories. Among these, three were academic schools, and two were vocational. One vocational school, being twice the size of the others, contributed approximately 200 students, while the remaining four schools, three academic and one vocational, contributed about 100 students each. This selection ensured that the final sample included approximately 300 students from each study program, while also facilitating organization and implementation.”
We appreciate your constructive feedback and hope this revision addresses your concern.
Comment 3:
-Please also add information on the method used to select the research group
Response 3:
We refer to our previous response to Comment 2 and hope this clarification adequately addresses your concern.
Comment 4:
-Please also add information on whether any questionnaires were rejected from the analysis and for what reason?
Response 4:
Thank you for your comment. We have added information on questionnaire exclusions. All consenting participants who submitted the questionnaire were included, except for seven who selected "Other" for gender identity. Due to the small sample size, meaningful statistical comparisons were not feasible, and including these responses could have affected the analysis robustness. To clarify this, we have added the following statement to subsection Statistical analysis:
L 191-195: "All participants who provided informed consent and completed the questionnaire were included in the dataset. However, seven responses were excluded from the analyses because the participants identified as 'Other' in response to the gender identity question (Boy, Girl, Other). Due to the small sample size in this category, meaningful statistical comparisons were not feasible."
Comment 5:
results: -Please highlight statistically significant results in tables
Response 5:
Thank you for your valuable suggestion. In response, we have now highlighted all statistically significant results in the regression table. Specifically, we have bolded all p-values below the significance threshold (p < 0.05) under the Sig. column to enhance readability and facilitate interpretation.
Comment 6:
-Please present key results for the study in graphical form, which will certainly be more attractive to potential recipients
Response 6:
Thank you for your valuable suggestion, which we agree will provide the reader with a clearer understanding of the results. Therefore, we have included Figure 1 presenting the standardized B-values with 95% CI error margins for the included predictors in a horizontal bar chart. We hope this meets your expectations (please see page 7).
Comment 7:
Discussion: -in this section, please take into account that well-being in the school environment among students , but not only, can be influenced by the attitude and approach as well as the implementation of physical activity.
Response 7:
Thank you for your valuable comment. We acknowledge that students' well-being in the school environment is influenced not only by individual experiences but also by broader institutional factors, including school policies, teacher attitudes, and the implementation of physical activity. However, we would like to clarify that this study is a baseline assessment, conducted prior to the physical activity intervention. As such, we do not yet evaluate the effects of implementation strategies but rather examine pre-existing factors associated with students' mental well-being.
However, to address your suggestion, we have expanded the discussion to highlight the broader role of school-based approaches to physical activity and well-being while noting the need for future research to explore the impact of structured interventions. The following addition has been made in the Discussion section:
L317-327: “It is important to acknowledge that wellbeing in the school environment is shaped not only by individual experiences but also by broader institutional factors, including the school’s approach to and implementation of physical activity (van Sluijs et al., 2021). While this study represents a baseline assessment conducted before the implementation of the physical activity intervention, prior research suggests that school policies, teacher attitudes, and the availability of structured physical activity opportunities significantly influence students’ engagement and wellbeing (Biddle and Asare, 2011; Guthold et al., 2020). Future research should examine how the implementation of structured physical activity programs impacts school climate and student wellbeing outcomes beyond individual perceptions of PE. These findings reinforce the critical role of well-structured PE programs in promoting mental wellbeing among students.”
Comment 8:
It should be emphasized that physical activity can be an excellent field for action to improve well-being and improve the quality of life among students.
Response 8:
Thank you for your insightful comment. We fully acknowledge the critical role of physical activity in promoting student well-being and quality of life. While this study does not assess the direct effects of a physical activity intervention, our findings suggest that perceived well-being in PE classes is a significant predictor of mental well-being, aligning with existing research on the benefits of physical activity.
To highlight the importance of physical activity as a key strategy for improving well-being, we have expanded the Discussion section as follows:
L 304-316: “Not surprisingly, the study demonstrated that perceived wellbeing in physical education (PE) classes significantly enhanced students’ overall mental wellbeing (b = .129; p = 0.002). This finding aligns with existing research indicating that physical education offers students crucial opportunities for physical activity, social interaction, and skill development, all of which are instrumental in enhancing self-esteem and mental health (Bailey et al., 2012; Janssen and LeBlanc, 2010). The beneficial effects of physical activity on mental health are well-documented, with evidence indicating improvements in mood, reduction in anxiety levels, and enhanced cognitive function (Biddle and Asare, 2011; Lubans et al., 2016). It should be emphasized that physical activity represents a powerful tool for enhancing well-being and improving students’ quality of life (Cadenas-Sanchez et al., 2021). While this study does not assess the direct effects of physical activity, future research should further explore how structured school-based interventions can optimize its implementation as a mental health-promoting strategy in educational settings.”
Reviewer 2 Report
Comments and Suggestions for Authors
The study addresses an important and timely topic—mental wellbeing among adolescents, particularly in the context of school climate. The focus on upper secondary students in Norway adds a unique cultural perspective. The use of validated scales (e.g., SWEMWBS, self-efficacy scale) and a robust statistical approach (stepwise multiple linear regression) enhances the credibility of the findings. The inclusion of gender and socioeconomic status (SES) as variables adds depth to the analysis, even though SES did not show significant effects. The findings have clear implications for educational practices, particularly in fostering supportive school environments to enhance mental wellbeing. Overall, the manuscript presents valuable insights into the association between school climate and mental wellbeing among Norwegian upper secondary students. However, addressing the following weaknesses and incorporating the suggested improvements would strengthen the study's contribution to the field and enhance its practical relevance for educators and policymakers:
1. The sample is limited to five schools in Eastern Norway, which may not be representative of the broader Norwegian population or other cultural contexts. The authors should acknowledge this limitation and suggest future research with a more diverse sample, including rural areas and other regions of Norway, to enhance generalizability.
2. While gender differences in mental wellbeing were noted, the discussion on why these differences exist and how they might be addressed is limited. The authors should expand on potential reasons for gender disparities (e.g., societal expectations, differential treatment in schools) and suggest targeted interventions to address these issues.
3. The study found no significant association between SES and mental wellbeing, which contrasts with existing literature. However, the discussion on why this might be the case is minimal. The authors should explore potential reasons for this finding, such as the relatively homogeneous SES of the sample or the influence of other mediating factors (e.g., school climate, peer relationships). They could also suggest future research with a more socioeconomically diverse sample.
4. The study provides valuable insights but does not explicitly discuss how these findings could inform educational policies or school-based interventions. The authors should include a section on policy implications, recommending specific strategies for schools to improve mental wellbeing (e.g., fostering self-efficacy, reducing social isolation, enhancing classroom climate).
5. The manuscript could benefit from a more explicit theoretical framework to guide the study. For example, the authors could draw on theories such as Self-Determination Theory (SDT) or Social Ecological Theory to explain how school climate influences mental wellbeing.
6. The authors should discuss how the Norwegian cultural context (e.g., emphasis on equality, welfare state) might influence the findings and how these results might differ in other cultural settings.
7. The authors should outline specific areas for future research, such as exploring the role of teacher-student relationships, the impact of extracurricular activities, or the influence of digital technologies on mental wellbeing.
Author Response
Response to Reviewer 2:
Overall comment:
The study addresses an important and timely topic—mental wellbeing among adolescents, particularly in the context of school climate. The focus on upper secondary students in Norway adds a unique cultural perspective. The use of validated scales (e.g., SWEMWBS, self-efficacy scale) and a robust statistical approach (stepwise multiple linear regression) enhances the credibility of the findings. The inclusion of gender and socioeconomic status (SES) as variables adds depth to the analysis, even though SES did not show significant effects. The findings have clear implications for educational practices, particularly in fostering supportive school environments to enhance mental wellbeing. Overall, the manuscript presents valuable insights into the association between school climate and mental wellbeing among Norwegian upper secondary students. However, addressing the following weaknesses and incorporating the suggested improvements would strengthen the study's contribution to the field and enhance its practical relevance for educators and policymakers:
Response:
Thank you for your thoughtful feedback and constructive suggestions, which we sincerely appreciate. We have carefully addressed your comments and made the necessary revisions, aiming to enhance the manuscript’s clarity and quality. We hope these adjustments adequately address your concerns.
Comment 1:
The sample is limited to five schools in Eastern Norway, which may not be representative of the broader Norwegian population or other cultural contexts. The authors should acknowledge this limitation and suggest future research with a more diverse sample, including rural areas and other regions of Norway, to enhance generalizability.
Response 1:
Thank you for your comment. We acknowledge that this may represent a potential limitation in terms of the study’s representativeness. To address this, we have added the following text under subsection Strengths and limitations:
L363-367: “Furthermore, while the five participating schools were geographically distributed across southeastern Norway, their location in mid-sized urban areas may limit the study’s representativeness. To improve external validity, future research should strive to include a more diverse sample, incorporating schools from various regions, including rural areas.”
Comment 2:
While gender differences in mental wellbeing were noted, the discussion on why these differences exist and how they might be addressed is limited. The authors should expand on potential reasons for gender disparities (e.g., societal expectations, differential treatment in schools) and suggest targeted interventions to address these issues.
Response 2:
Thank you for your insightful feedback. We recognize that our discussion on potential explanations for the lower mental well-being reported by girls may have lacked clarity, particularly in lines 274–281, where we addressed this issue in relation to classroom climate. In that section, we clarified that the observed gender difference was not driven by classroom climate. However, our findings indicate that self-efficacy, social isolation, and well-being in physical education significantly contributed to this disparity, suggesting that girls may generally experience lower self-efficacy, greater social isolation, and/or more negative experiences in physical education.
In response to your comment, we have revised the manuscript to enhance clarity and have incorporated the following addition to the discussion:
L282-290: “While our study did not directly examine body image satisfaction, social comparison, or academic stress, prior research has highlighted that adolescent girls often report lower body image satisfaction, greater vulnerability to social comparison, and higher academic stress related to societal expectations, factors that have been linked to self-esteem and overall wellbeing (Barene et al., 2022; Kaczmarek and Trambacz-Oleszak, 2021; Papageorgiou et al., 2022). These broader patterns may help contextualize the gender differences observed in our study, underscoring the need for further research into the complex interplay of psychological and social factors in adolescent mental health”.
Comment 3:
The study found no significant association between SES and mental wellbeing, which contrasts with existing literature. However, the discussion on why this might be the case is minimal. The authors should explore potential reasons for this finding, such as the relatively homogeneous SES of the sample or the influence of other mediating factors (e.g., school climate, peer relationships). They could also suggest future research with a more socioeconomically diverse sample.
Response 3:
Thank you for your comment. While we had briefly addressed some of the factors you mentioned, we have expanded the discussion on the absence of associations between SES and mental wellbeing. We hope this revision sufficiently addresses your concerns. The updated section now reads as follows:
L328-346: “Unlike previous studies identifying SES as a key predictor of mental wellbeing (Pillas et al., 2014; Reiss, 2013), the significant association observed in Model 1 (p = 0.001) disappeared after adjusting for Spending time with friends and Health satisfaction in Model 2 (p = 0.068) and was further diminished in Model 3 (p = 0.497). This suggests that social relationships and perceived health mediate the SES-wellbeing link, consistent with research highlighting their protective role (Coyle et al., 2022). Another possible explanation is the limited socioeconomic diversity in our sample, which may have reduced variability and obscured potential associations. Additionally, the findings align with the resource substitution hypothesis (Schoon et al., 2021; Shi et al., 2023), which posits that strong social and environmental support can mitigate the negative effects of SES disparities. In this context, school-related factors, such as supportive peer and teacher relationships, likely played a compensatory role, buffering the impact of SES on wellbeing (Patalay and Fitzsimons, 2018). These protective mechanisms may have weakened the direct association between SES and mental wellbeing, underscoring the importance of social resources in shaping adolescent mental health outcomes. Future research should examine these dynamics in more socioeconomically diverse samples and employ longitudinal designs to clarify how school environments and peer relationships mediate SES effects over time. This would help identify targeted interventions to reduce socioeconomic disparities in adolescent mental health.”
Comment 4:
The study provides valuable insights but does not explicitly discuss how these findings could inform educational policies or school-based interventions. The authors should include a section on policy implications, recommending specific strategies for schools to improve mental wellbeing (e.g., fostering self-efficacy, reducing social isolation, enhancing classroom climate).
Response 4:
We agree that adding a section on practical implications and recommendations strengthens the discussion. Accordingly, we have included the following paragraph at the end of the discussion section:
L368-389: “Educational strategies and future research directions
Our findings underscore the importance of school climate in shaping students' mental wellbeing. Self-Determination Theory (SDT) highlights that when students experience autonomy, competence, and relatedness, they are more likely to develop intrinsic motivation, emotional resilience, and overall wellbeing (Ryan and Deci, 2000). The results indicate that positive teacher-student relationships, peer support, and engagement in school activities contribute to students' sense of belonging and self-efficacy, supporting SDT’s emphasis on fulfilling these psychological needs. Similarly, the Socio-Ecological Model (SEM) illustrates how school climate interacts with individual and social factors to shape wellbeing (Özdogru, 2011). The classroom environment, peer relationships, and extracurricular participation play a crucial role in either mitigating or reinforcing SES-related disparities, social isolation, and differences in health satisfaction. Our findings suggest that a supportive school climate can buffer the negative effects of socioeconomic disadvantages by fostering positive interactions, meaningful engagement, and emotional security. These results highlight the need for school-based interventions that enhance peer integration, improve teacher-student relationships, and create inclusive learning environments.
Future research should further explore these mechanisms by examining how teacher-student dynamics contribute to self-efficacy, the impact of extracurricular activities on social connectedness, and the role of digital technology as both a risk factor and a support tool for mental wellbeing. Understanding these factors will provide valuable insights for designing evidence-based educational policies that promote student wellbeing in diverse learning contexts.”
Comment 5:
The manuscript could benefit from a more explicit theoretical framework to guide the study. For example, the authors could draw on theories such as Self-Determination Theory (SDT) or Social Ecological Theory to explain how school climate influences mental wellbeing.
Response 5:
Thank you for your insightful comment. We agree that integrating the mentioned theoretical frameworks into the introduction will strengthen the study’s rationale. Accordingly, in the revised version of the introduction, we have included the following paragraph, which we hope addresses your feedback:
L34-44: “School climate influences students’ mental wellbeing by shaping their social interactions, sense of belonging, and academic motivation. Self-Determination Theory (SDT) highlights the importance of autonomy, competence, and relatedness in fostering wellbeing (Ryan and Deci, 2000). When students feel supported by teachers and peers, experience meaningful learning, and develop self-efficacy, they are more likely to engage in school positively. Conversely, a lack of support or inclusion can undermine motivation and wellbeing. The Socio-Ecological Model (SEM) further explains how individual wellbeing is influenced by interactions across multiple environmental levels (Özdogru, 2011). Supportive peer networks, inclusive extracurricular activities, and school-based mental health initiatives create an environment that fosters a sense of belonging and emotional security (Darling-Hammond and Cook-Harvey, 2018).”
Comment 6:
The authors should discuss how the Norwegian cultural context (e.g., emphasis on equality, welfare state) might influence the findings and how these results might differ in other cultural settings.
Response 6:
Thank you for your comment. In the revised version of the discussion, we have included the following description:
L347-356: “The findings should be understood in the Norwegian context, where equal access to education, healthcare, and social support most likely helps reduce SES differences in mental wellbeing. Instead, self-efficacy emerged as the strongest positive predictor, followed by time spent with friends and health satisfaction, emphasizing the role of personal agency and social connectedness. Additionally, classroom climate ranked third, reinforcing the impact of inclusive, student-centered learning environments. In contrast, social isolation was a significant negative predictor, underscoring its harmful effects. These results suggest that in Norway, school climate and peer relationships are central to adolescent mental wellbeing, differing from societies where SES plays a stronger role due to greater economic inequality or competitive education systems (Reiss, 2013; Rudolf and Lee, 2023).”
Comment 7:
The authors should outline specific areas for future research, such as exploring the role of teacher-student relationships, the impact of extracurricular activities, or the influence of digital technologies on mental wellbeing.
Response 7:
Thank you for your comment, which we fully agree with. In the revised manuscript, we have incorporated this information and referenced the subsection Educational strategies and future research directions, as addressed in Response 4, which we hope satisfactorily addresses your concern.
Round 2
Reviewer 2 Report
Comments and Suggestions for Authors
No further comments.
Comments on the Quality of English LanguageNo further comments.